# Gene Polymorphism of Biotransformation Enzymes and Ciprofloxacin Pharmacokinetics in Pediatric Patients with Cystic Fibrosis

**DOI:** 10.3390/biomedicines10051050

**Published:** 2022-05-02

**Authors:** Sergey K. Zyryanov, Elena A. Ushkalova, Elena I. Kondratyeva, Olga I. Butranova, Yulia A. Kondakova

**Affiliations:** 1Peoples’ Friendship University of Russia (RUDN University), 6 Miklukho-Maklaya Street, Moscow 117198, Russian Federation; zyryanov-sk@rudn.ru (S.K.Z.); ushkalova_ea@pfur.ru (E.A.U.); 2Department of Health of the City of Moscow, City Clinical Hospital No. 24, State Budgetary Institution of the City of Moscow, Pistzovaya Str. 10, 127015 Moscow, Russia; 3Research Centre for Medical Genetics, 1 Moskvorechyie Str., 115522 Moscow, Russia; elenafpk@mail.ru; 4GBUZ NSO City Children’s Clinical Hospital of Emergency Medical Aid, Krasny Prospekt 3, 630007 Novosibirsk, Russia; yulakondakova@rambler.ru

**Keywords:** cystic fibrosis, children, ciprofloxacin, pharmacokinetics, biotransformation enzymes, genetic polymorphism

## Abstract

(1) Background: Ciprofloxacin (CPF) is widely used for the treatment of cystic fibrosis, including pediatric patients, but its pharmacokinetics is poorly studied in this population. Optimal CPF dosing in pediatric patients may be affected by gene polymorphism of the enzymes involved in its biotransformation. (2) Materials and Methods: a two-center prospective non-randomized study of CPF pharmacokinetics with sequential enrollment of patients (n-33, mean age 9.03 years, male-33.36%), over a period from 2016 to 2021. All patients received tablets of the original CPF drug Cyprobay^®^ at a dose of 16.5 mg/kg to 28.80 mg/kg. Blood sampling schedule: 0 (before taking the drug), 1.5 h; 3.0 h; 4.5 h; 6.0 h; 7.5 h after the first dosing. CPF serum concentrations were analyzed by high performance liquid chromatography mass spectrometry. The genotype of biotransformation enzymes was studied using total DNA isolated from whole blood leukocytes by the standard method. (4) Results: a possible relationship between the CA genotype of the CYP2C9 gene (c.1075A > C), the GG genotype of the CYP2D6*4 gene (1846G > A), the AG genotype of the GSTP1 gene (c.313A > G), the GCLC* genotype 7/7 and the CPF concentration in plasma (increased value of the area under the concentration–time curve) was established. Conclusions: Gene polymorphism of biotransformation enzymes may affect ciprofloxacin pharmacokinetics in children.

## 1. Introduction

Cystic fibrosis (CF) is one of the most common autosomal recessive diseases in clinical practice. In European countries, the incidence of CF is 1/3000 to 1/6000 [1], in the Russian Federation it varies from 1/2500 to 1/17,000 [2], while screening data detect this pathology in newborns in 1/10,000 [2]. CF is caused by mutations in the cystic fibrosis transmembrane conductance regulator (*CFTR*) gene [1]. About 2000 mutations of *CFTR* gene have been described [3]. Dysfunction of *CFTR* leads to impaired hydration of epithelial surfaces and to the formation of a thick, dehydrated secretion in the excretory glands. The main lesions are seen in the lungs and pancreas; changes are also observed in many other organs and systems, including the intestines, liver and gonads [4]. Multiple lesions of organs and systems in patients with CF may lead to changes affecting drugs pharmacokinetics.

The absorption rate of drugs is determined by the morpho-functional state of the gastrointestinal tract (GIT). *CFTR* is expressed at low levels in the stomach, its maximal distribution is observed in the intestine [5]. Expression has a cranial-caudal gradient: the highest concentration is observed in the duodenum, the lowest-in the ileum [5]. *CFTR* defect in the intestine leads to a disruption of bicarbonate production and a decrease in mucus hydration, resulting in an increase in acidity in the upper small intestine, an increase in the viscosity of the mucous secretion, thickening of the mucous membrane and impaired GIT motility [5]. Clinical manifestations include meconium ileus, constipation, distal intestinal obstruction syndrome, gastroesophageal reflux disease and intestinal microflora disorders [6]. A decrease in intestinal pH in patients with CF may affect the absorption and bioavailability of the drugs that require an alkaline environment for complete dissolution [7].

Changes in pancreatic function lead to the development of exocrine insufficiency, resulting in malabsorption, diarrhea and steatorrhea, which contribute to the decrease in fats and fat-soluble drugs and vitamins absorption [8,9,10]. *CFTR* defect can also contribute to the development of pancreatic tissue fibrosis followed by a decrease in insulin secretion and to the development of CF-related diabetes [11]. 

Liver damage in CF can affect the volume of drugs distribution and metabolism. CF is accompanied by dysfunction of the hepatobiliary ducts, since *CFTR* is located precisely on the apical surface of cholangiocytes [12]. As a result, focal biliary cirrhosis, multilobular cirrhosis and cholangiopathy are formed. Intrahepatic cholestasis is also typical for CF patients, resulting in alteration of drug absorption [13]. Cirrhosis may contribute to a decrease in CYP450 system enzymes expression, impaired hepatic blood flow, a decrease in the concentration of albumin and α1-acid glycoprotein in blood plasma [14]. There may also be some increase in the activity of CYP1A2 and CYP2C8, glucuronyltransferase, acetyltransferase and sulfotransferase, which can lead to an acceleration of the drugs’ final elimination [15]. 

Kidney injury in CF is not widespread: according to Yahiaoui Y et al. (2009), only 2.5% of patients required a kidney biopsy due to the disease [16]. According to Santoro D (2017), proteinuria was found in 11.7% of patients with cystic fibrosis and chronic renal failure in 14.28% [17]. Lai S (2019) discovered a decrease in glomerular filtration rate <90 mL/min/1.73 m^2^ in 28.8% of patients with CF [18]. Established changes in kidney function may affect the rate of drugs excretion in CF, especially against the background of a long course of the disease and the need for lung transplantation [19]. 

Pharmacotherapy of CF is primarily aimed at the cessation of bronchopulmonary manifestations and includes prescribing of the following pharmacological groups: mucoactive drugs (dornase alpha, hypertonic solution, bronchodilators, mannitol), the action of which is aimed at thinning and facilitating of secretions evacuation; drugs aimed at regulation of the inflammation (glucocorticoids, non-steroidal anti-inflammatory drugs and antimicrobials) [19]. The role of antibacterial drugs in CF can hardly be overestimated: according to the Russian Register of Patients with CF for 2019, the frequency of inhaled antibiotics (AB) use in children was 44.63% of intravenous AB-26.49%, and of oral AB-47.07%, respectively [20]. Against the background of the widespread prevalence of *P. aeruginosa* infection in patients with CF [21] in the Russian Federation, CPF is the drug of choice for oral administration, and it is included in the scheme for the eradication of *P. aeruginosa* and the treatment of “light” exacerbations of the bronchopulmonary process at a dose of 30–40 mg/kg/day in combination with inhaled antibiotics. According to the official instruction, the use of CPF (Ciprobay^®^) is approved for children with CF aged 5 to 17 years with confirmed *P. aeruginosa* infection. In real clinical practice, prescription is observed in patients younger than 5 years. Despite the expansion of indications for the use of CPF in pediatrics and the decrease in the age limit for prescribing, there is limited information on the pharmacokinetics which does not allow developing a clear therapeutic algorithm for the young category of patients [22,23,24,25,26]. Studying pharmacokinetics of CPF in children of different ages with parallel estimation of the effects of the genotype of phase 1 and 2 xenobiotic biotransformation enzymes will provide us with data essential for the safe and effective antibiotic therapy. 

We aim to study CPF pharmacokinetic parameters in pediatric patients with CF and their relationship with the genotype of phase 1 and 2 xenobiotic biotransformation enzymes.

## 2. Materials and Methods

Study design: a two-center (clinical centers of the Moscow and of the Siberian Federal District of the Russian Federation) prospective non-randomized study of CPF pharmacokinetic profile with sequential enrollment of patients. The study population included 33 hospitalized patients with CF aged 2 to 18 years (12 boys and 21 girls) enrolled from 2016 to 2021. The study was performed in accordance with the legal requirements and the Declaration of Helsinki and with present Eurasian economic union guidelines for good clinical practice. Written consent was obtained from parents (or legal guardians). The inclusion criteria: the presence of dated and signed informed consent, the age of patients from 2 to 18 years, the diagnosis of CF based on international and Russian guidelines. Exclusion criteria from the study were: drug allergy to CPF, severe exacerbation of CF, acute and chronic cardiovascular failure, renal failure, liver cirrhosis in the decompensation stage.

All patients received tablets of the original CPF drug Cyprobay^®^ (Bayer Schering Pharma AG, Germany) at a dose of 16.5 mg/kg to 28.80 mg/kg. Blood sampling schedule: 0 (before taking the drug), 1.5 h; 3.0 h; 4.5 h; 6.0 h; 7.5 h after the first dosing. Analysis of pharmacokinetic parameters included data only of those patients who received a dose of the drug and donated at least 75% of blood samples.

CPF serum concentrations were analyzed by high performance liquid chromatography on a Shimadzu LC-20 AD Series chromatograph equipped with a Shimadzu LC-MS 8030 triple quadrupole mass detector (Shimadzu, Tokyo, Japan). The analytical accuracy of the determination of CPF was 5.3% (0.1 μg/mL) at the level of LLOQ (lower limit of quantitation) and 2.1% (5 μg/mL) at the level of ULOQ (upper limit of quantitation).

Statistical data processing was carried out using IBM^®^ SPSS^®^ Statistics Version 21.0 software (IBM Corp. Released 2012. IBM SPSS Statistics for Windows, Version 21.0. Armonk, NY, USA: IBM Corp.). Descriptive statistics for demographic and pharmacokinetic parameters included mean, standard deviation (SD), median, minimum and maximum values and number of valid cases (N).

The following pharmacokinetic parameters were assessed (using a non-compartmental method according to the recorded values during the observation period of 7.5 h and 12 h after taking the drug):Maximum concentration (Cmax) for a period of 7.5 h after dosing.Time to reach maximum concentration (Tmax) over a period of 7.5 h and 12 h after dosing.The area under the concentration–time curve after taking the drug until the last measurement is above the limit of quantification for the observation period of 7.5 h and 12 h (AUC0-t).

To compare pharmacokinetic parameters, individual values of Cmax and AUC were normalized according to the dose received (mg/kg): AUC0-t_norm and Cmax_norm. The effect of age on the values of the pharmacokinetic parameters AUC0-t_norm and Cmax_norm was analyzed using an ANOVA after log transformation. Differences were considered statistically significant at *p* < 0.05.

The genetic study included the search for pathogenic genetic variants of CF using data from regional genetic centers and the Moscow State Scientific Center. The name of the mutations was entered, respectively, in the international database *CFTR1* and *CFTR2* [27]. The allelic frequency of all detected mutations and the severity of the genotype were determined. The “severe” genotype was determined in the presence of class I–III mutations on two chromosomes; “mild” genotype-in the case of the presence of at least one mutation of class IV-V. Information about the *CFTR* genotype of Russian patients was taken from the Russian Register of Patients with CF 2019, the archives of the Laboratory of Genetic Epidemiology of the Federal State Budgetary Scientific Institution “Moscow State Scientific Center” and the Research Institute of Medical Genetics in Tomsk.

Data on the genotype of biotransformation enzymes of phases 1 and 2 xenobiotics were based on the analysis of total DNA isolated from whole blood leukocytes by the standard method. The patients’ genomic DNA was isolated from venous blood leukocytes using the Wizard Genomic DNA Purification Kit (Promega, Madison, WI, USA) in accordance with the manufacturer’s recommendations.

The analysis of polymorphic variants of xenobiotics biotransformation enzyme genes was carried out by PCR followed by restriction by specific endonucleases (RFLP), or by direct analysis of the amplification fragment length polymorphism (AFLP) during separation by gel electrophoresis. Genes and polymorphic variants selected for study:

I phase: CYP2C9*3 (rs1057910; c.1075A > C; I359L), CYP2C9*2 (rs1799853; c.430C > T; R144C), CYP2C19*2 (rs4244285; c.681G > A), CYP2C19*3 (rs4986893; c.636G > A; W212X), CYP2D6*4 (rs3892097; 1846G > A), CYP3A4*3 (rs4986910; M445T; c.1334T > C), CYP3A4*1B (rs2740574; c.-392C > T);

II phase: deletion polymorphism of genes GSTT1 and GSTM1; GSTP1 (c.313A > G); GCLC (TNR(GAG), c.-129C > T); GCLM (c.-588C > T); NAT2 (c.191G > A, c.282C > T, c.341T > C, c.434A > C, c.481C > T, c.590G > A, c.803A > G, c.845A > C, c.857G > A).

Statistical analysis of the effect of age on the values of AUC0-t_norm and Cmax_norm was performed using analysis of variance after a logarithmic transformation. Differences were considered statistically significant at *p* < 0.05.

The analysis of the relationship between the pharmacokinetic parameters of CPF and the genotypes of phase 1 and 2 xenobiotic biotransformation enzymes was carried out using the statistical packages SAS 9.4, STATISTICA 12 (SAS Institute Inc., Cary, NC, USA) and IBM-SPSS-21. The critical value of the level of statistical significance when testing null hypotheses was taken equal to 0.05 or 0.01. For all quantitative traits in the compared groups, the arithmetic means and root mean square (standard) errors of the means, coefficient of variation, median and quartiles were evaluated. To compare the distributions of parameters of quantitative traits nonparametric methods were used: Kruskal-Wallis analysis of variance with Wilcoxon Rank-Sum Test, the Van der Waerden test and the median test. The study of correlations between pairs of quantitative traits was carried out using the Pearson and Spearman correlations. In the study of the multidimensional dependence of grouping features on a set of quantitative features, the method of discriminant analysis was used. We used the method of logistic regression with stepwise algorithms for including and excluding predictors.

Logistic regression included analysis of next predictors:
(1)Clinical and demographic
-Age,-Sex,-Anthropometric data,-Fecal elastase concentration,-Chronic infection with *P. aeruginosa*,-Presence of meconium ileus,-Presence of genotype F508del/F508del,-Liver damage (estimated with liver enzymes elevation and ultrasound diagnostics),-Spirometry (FEV1, FVC).(2)Pharmacokinetic parameters.

## 3. Results

### 3.1. Baseline Characteristics of Study Population

The mean age of the patients was 9.03 years, the median was 9.0. According to the recommendations of the FDA Guidance (1998) Food and Drug Administration [28] and the recommendations of the British National Formulary for Children (2011–2012) [29] patients were divided into three age groups: 2 to 6 years (*n* = 8); 6 to 12 years (*n* = 14); 12 to 18 years (*n* = 11). The basic characteristics of the study population are given in Table 1.

### 3.2. Pharmacokinetic Parameters of CPF in Children with CF

The lowest values of the mean AUC0-t, µg·h/mL and AUC0-t_norm, (µg·h/mL)/(mg/kg) (69.84 ± 35.29 and 3.33 ± 1.98, respectively) were defined in the group of children aged 2–6 years. The highest values of the mean AUC0-t, µg·h/mL and AUC0-t_norm, (µg·h/mL)/(mg/kg) (100.53 ± 46.6 and 4.77 ± 2.20, respectively) were obtained in the group of 12–18 years. A similar upward trend in the mean values seen with increased age was also observed for the values of Cmax (µg/mL) and Cmax_norm (µg/mL)/(mg/kg). In the younger age group, the mean values were 22.32 ± 13.15 µg/mL and 1.03 ± 0.55 (µg/mL)/(mg/kg), respectively, in the older age group the mean values were 23.63 ± 9.51 µg/mL and 1.12 ± 0.47 (µg/mL)/(mg/kg), respectively. The obtained pharmacokinetic parameters are shown in Table 2.

Although AUC values were calculated only for a period of 7.5 h after dosing, they can be considered to some extent as the reciprocal of the total clearance of the drug. According to the AUC values, one can indirectly judge the trend in the dependence of the total clearance of CPF on age. Available data suggest that, on average, the apparent total clearance of CPF decreases with age. The median time to reach the maximum plasma concentration of CPF in the group of children aged 2–6 years was 1.5 h (1.5–3.0 h), in the groups 6–12 years and 12–18 years, the median was estimated as 3.0 h. The results revealed high inter-individual variability in all studied pharmacokinetic parameters. At the same time, the highest coefficients of variation (CV) were observed in the group of patients aged 2–6 years, except for Tmax. Figure 1 below shows graphs of the dynamics of the concentration of CPF, measured within 7.5 h after a single dose of the drug in different age groups.

A visual assessment of individual graphs of changes in the concentration of CPF in children of different age groups revealed a trend towards higher values of the maximum concentration of the drug in children aged 6 years and older, compared with a children aged 2–6 years. The results obtained may indirectly indicate a higher intensity of the total clearance of the drug in children of the younger age. In the same age group, taking into account the data on the minimum inhibitory concentration (MIC) for *P. aeruginosa* ≥ 1 μg/mL, a single dose of CPF at an average dose of 20.1 ± 0.73 mg/kg did not achieve the ratio Cmax/MIC > 10 in blood plasma, which is a necessary for fluoroquinolones to provide eradication of the pathogen.

### 3.3. Study of the Effect of the Genotype of Xenobiotic Biotransformation Enzymes of the 1st and 2nd Phases on the Pharmacokinetic Parameters of CPF in Children and Adolescents

The number of patients with estimated genotypes was 32 out of 33 (analysis of one patient failed due to technical reasons). Table 3 contains characteristics of the study population corresponding to different genetic variants of the CYP450 family of monooxygenases, glutathione-S-transferases (GST) and N-acetyltransferases (NAT).

The parameters of the physical development of patients with different variants of the CYP2C9*3 gene polymorphism (I359L, c.1075A > C) are given in the Appendix A (Appendix A). Table 4 contains the pharmacokinetic parameters of CPF, presented depending on CYP2C9*3 gene polymorphism (I359L, c.1075A > C).

Data from a comparative analysis of group mean values of CPF concentration at point 5 for the grouping trait “CYP2C9*3 I359L(c.1075A > C) polymorphism” in patients with AA, CA, CC genotypes are shown in Table 5.

Discriminant analysis to identify the relationship between the dependent variable “CYP2C9*3 gene polymorphism I359L(c.1075A > C)” and the pharmacokinetic parameters of CPF in patients with AA and CA genotypes included the selection of two significant predictors: the concentration of CPF at point 3 and at point 5. Achieved levels the significance for these features did not exceed 0.03. The result of the discriminant analysis is presented in Table 6.

The average percentage of correct reclassification was 96.67%. It follows from the theory of discriminant analysis that the higher the percentage of correct reclassification, the better the features included in the discriminant functions explain the difference between the compared groups. The percentage of correct reclassification was highest in patients with the AA genotype (100%). The results of assessing the presence of multivariate relationships between one grouping variable “CYP2C9*3 I359L(c.1075A > C) polymorphism” and a set of other qualitative (grouping) and quantitative signs using the logistic regression method allow us to establish a subset of predictors that can affect the probability of attributing a particular observation into one of the analyzed groups.

Evaluation of parameters in patients with different genotype variants (AA, CA, CC) of the CYP2C9 I359L gene (c.1075A > C) revealed association between CA genotype and higher values of CPF concentration at point 5 compared with other genotypes (percent concordant = 91.3% and the D-Somer coefficient (Somers’ D) = 0.826)

Statistical analysis carried out using logistic regression with the exclusion of the sign “CPF concentration at sampling points 1, 2, 3, 5” allowed us to obtain the result presented in Table 7, here the percentage of concordance was 92.5%, the D-Somer coefficient was 0.856.

When comparing the magnitudes of the modules of standardized regression coefficients, pharmacokinetic predictors (Cmax, µg/mL and AUC, µg·h/mL) were identified, which were most closely associated with the dependent trait “CYP2C9*3 gene polymorphism I359L (c.1075A > C)”. The presence of the CA and CC genotypes was associated with high AUC values, µg·h/mL compared with the AA group. The same groups were characterized by lower values of Cmax, µg/mL. The predictor “total dose of CPF” was also used as a predictor in the resulting equation, but with the lowest modulus. In the CA group, patients received a higher total dose of CPF. The absence of chronic staphylococcal infection was associated with the CA genotype. The predictor “chronic *P. aeruginosa* infection” did not find a relationship with the AA, CA and CC genotypes.

Parameters of physical development in groups of patients with different genotypes of CYP2D6*4 gene polymorphism (1846G > A) are given in the Appendix A. Table 8 provides information on the pharmacokinetic parameters of CPF in patients with various genotypes.

A high variability was obtained in each of the three groups for all pharmacokinetic parameters. Means and medians had no clear differences. When testing hypotheses about the equality of group means and variances for each of the quantitative characteristics using one-way analysis of variance (ANOVA) and evaluating the non-parametric tests of Van der Waerden, the median test, Kruskal-Wallis, Siegel-Tukey test and Ansari-Bradley test, when comparing the distributions of these quantitative parameters separately in subgroups GG, GA and AA formed by the qualitative trait “CYP2D6*4 (1846G > A) polymorphism” no differences were found. When conducting a discriminant analysis, the relationship between the dependent variable “CYP2D6 * 4 (1846G > A) polymorphism” and groups with variants of the GG, GA and AA genotypes with the previously indicated quantitative traits was not revealed. When building a logistic regression model with a step-by-step algorithm for including and excluding predictors, a logistic regression equation with a concordance percentage of 100% and a D-Somer coefficient of 1.000 was selected, results are shown in the Table 9.

The GG group included older patients, which explains the higher parameters of physical development in the GG group compared to the GA and AA groups. The relationship between the GG genotype and meconium ileus in anamnesis was revealed. Liver damage occurred at a higher frequency in the group of GA and AA genotypes. The values of the area under the concentration–time curve were higher in the GG group, which was comparable to the age characteristics of the group. High Cmax_norm values per mg/kg were correlated with AG and AA genotypes.

When testing hypotheses about the equality of group means and variances for each of the quantitative traits, no differences were found in groups without deletion and “Zero” genotype formed by the qualitative trait GSTM1. Furthermore, no differences were found in the groups of the NN and DD genotypes of the GSTT1 gene.

When conducting a discriminant analysis, the relationship between the dependent variable “GSTM1” and the groups “without deletion” and “Zero genotype” with the previously indicated quantitative traits was not revealed.

Using the method of logistic regression, the relationship between the dependent variable “GSTP1 gene polymorphism” and a number of quantitative and qualitative traits was studied. No relationship between the pharmacokinetic parameters of CPF and the genotypes of the GSTP1 gene polymorphism was found during the analysis by the method of logistic regression with a dependent parameter. Discriminant analysis and analysis using the logistic regression method for the dependent variable “GSTT1” did not find a relationship between the pharmacokinetic parameters of CPF and the NN and DD genotypes. Parameters of physical development of children with different genotypes of GSTP1 gene polymorphism are shown in the Appendix A. Pharmacokinetic parameters of CPF in children depending on different types of the GSTP1 gene polymorphism are presented below in the Table 10.

The analysis revealed a high variability of all studied pharmacokinetic parameters. When testing hypotheses about the equality of group means and variances for each of the quantitative traits in the groups of genotypes AA, AG and GG, formed by the qualitative trait “GSTP1 gene polymorphism”, no differences were found. When conducting a discriminant analysis, the relationship between the dependent variable “GSTP1 gene polymorphism” in groups AA, AG and GG with pharmacokinetic parameters was not revealed. Using the method of logistic regression, the relationship between the dependent variable: “GSTP1 gene polymorphism” and a number of quantitative and qualitative traits was studied. Table 11 shows the performance of the logistic regression equation, for which the percentage of concordance was 100%, the D-Somer coefficient—1.000.

This equation included several predictors. In Table 12, the predictors are arranged in descending order of the values of the module of standardized regression coefficients (Standardized Estimate), the most significant predictors have larger values of the module of standardized regression coefficients.

Considering that patients received CPF at a dose of 16.5 mg/kg to 28.8 mg/kg, the parameters normalized to the received dose were introduced—Cmax_norm, µg/mL and AUC0-t_norm, (µg·h/mL)/(mg/kg). Table 13 below shows the identified relationships between pharmacokinetic parameters and clinical signs and genotypes of the GSTP1 gene.

Parameters of the physical development of children with different variants of the TNR(GAG) polymorphism genotype of the GCLC gene are presented in the Appendix A. The pharmacokinetic parameters of CPF in children and adolescents with different variants of the GCLC gene are shown below in Table 14.

The results of a discriminant analysis of the relationship between the dependent variable GCLC, groups with genotype variants 7/7, other genotypes and the previously indicated quantitative traits revealed only one significant predictor—the concentration of CPF 7.5 h after taking the drug.

Using the method of logistic regression, the relationship between the dependent variable GCLC and a number of quantitative and qualitative traits was studied. Table 15 shows the indicators of the logistic regression equation, reflecting the strength of the relationship between the actual belonging of the observations to the specified groups. For this equation, the percentage of agreement concordance was 88%, the D-Somer coefficient was 0.764.

Analysis of the relationship between the studied parameters, presented in the table, show that the GCLC 7/7 genotype was associated with the concentration of CPF at point 5 (7.5 h after the start of the drug). As the concentration of CPF at point 5 increases, the probability of being assigned to the group with the 7/7 genotype increases, in addition, the presence of staphylococcal infection also determines the assignment of the patient to this group. When excluding concentrations at sampling points from the number of predictors, an equation was obtained for which the percentage of agreement concordance is 91.1%, the D-Somer coefficient was 0.821. The final form of the equation is presented in Table 16.

According to the module-standardized regression coefficients, the most significant predictors were chronic staphylococcal infection and Tmax. The probability of assigning a patient to a group with genotype 7/7 is higher in the presence of chronic staphylococcal infection, prolongation of Tmax, increase in weight values (no direct correlation with height). It is likely that the group with genotype 7/7 included more patients from the older age group or patients with better nutritional status.

When testing hypotheses about the equality of group means and variances for each of the quantitative characteristics using a one-way analysis of variance (ANOVA) and evaluating the non-parametric tests of Van der Waerden, the median test, Kruskal-Wallis, Siegel-Tukey test and Ansari-Bradley test, when comparing the distributions of these quantitative indicators separately in subgroups formed by qualitative signs NAT2(341T > C), NAT2(481C > T), NAT2(803A > G), NAT2(282C > T) and NAT2(590G > A), no differences between subgroups were found. Analysis of the relationship between one qualitative trait, acting as a dependent, resulting indicator, and a subset of quantitative and qualitative traits was carried out using a logistic regression model with a step-by-step algorithm for including and excluding predictors. Appendix A includes indicators of physical development in children with various variants of the NAT2(341T > C) genotypes. Pharmacokinetic parameters of CPF in children with different NAT2(341T > C) genotype variants are also presented in the Appendix A. In the course of the analysis by the method of logistic regression, an equation with a high percentage of agreement, concordance equal to 80.2% and a D-Somer coefficient of 0.604 was selected. The results are shown in Table 17.

Several predictors entered the equation at once, the most significant included: Cmax, AUC0-t, AUC0-t_norm, Cmax_norm. A relationship was found between high values of AUC0-t, Cmax_norm, longer Tmax and heterozygous carriage of the NAT2 genetic variation (341T > C). The medians and means of pharmacokinetic parameters were practically comparable in the TT and SS groups. Medians in groups with TT, TC and CC genotypes were: Cmax-17.89; 26.20; 23.20 and Cmax_norm-0.94; 1.26; 1.13, respectively. Medians of AUC0-7.5 h and AUC0-7.5 h_norm in patients with the TT genotype were 74.56 and 3.31; TS-100.75 and 4.16; SS-61.79 and 3.09, respectively. The highest median value in the group of the TC genotype may be associated with the age structure of the groups. In the CC group there was the largest number of patients of younger patients (2 to 5 years) and the mean age in this group was 6.83 ± 3.66 years, compared with the groups TT (9.0 ± 3.40 years) and CT (10, 13 ± 4.78 years). An analysis was carried out by the method of logistic regression without including quantitative characteristics of a group of children aged 2 to 5 years. The equation with a concordance percentage of 88.3% and a D-Somer coefficient of 0.765 also included several predictors at once. The parameters of this equation are shown in the Appendix A. According to the equation, the carriage of “slow” alleles of TS and SS was interconnected with the highest values of CPF concentrations after 1.5 h, 3 h, 7.5 h of taking the drug, with Cmax_norm, and longer Tmax. The higher the received dose of CPF (mg/kg), the higher the probability of assigning a patient to a group with TC and CC genotypes.

A statistical analysis was carried out in order to identify the relationship between pharmacokinetics of CPF and some clinical parameters of the disease in a particular patient with the F508del/F508del genotype. General characteristics for homozygotes for the genetic variant F508del and for “other” genotypes are given in Appendix A, and pharmacokinetic parameters of CPF for the same population are given in Appendix A.

When testing hypotheses about the equality of group means and variances for each of the 15 quantitative traits using a one-way analysis of variance (ANOVA) and non-parametric tests of Van der Waerden, median test, Kruskal-Wallis, Sigel-Tukey and Ansari-Bradley, when comparing distributions of quantitative traits separately in subgroups “F508del/F508del genotype” and “other genotypes”, no differences between groups were found. Using the method of discriminant analysis, the relationship between the indicated genotypes and qualitative and quantitative traits was not revealed. Table 18 below shows the estimation of logistic regression parameters in patients with the F508del/F508del genotype and other genotypes.

The coefficient of agreement for this equation was 90.6%, D-Somer coefficient-0.813. Several predictors entered the equation at once. A relationship was found between the F508del/F508del genotype and chronic persistent Pseudomonas aeruginosa infection, high concentrations of CPF after 1.5, 3 and 7.5 h of taking the drug, and increased values of Cmax_norm. It should be noted that the dose of CPF calculated per kg of body weight was also a significant predictor, and a higher dose of CPF was received in the group of homozygotes according to the F508del variant. The most significant predictor according to the module-standardized regression coefficient was Cmax, high values were interconnected with “other” genotypes. The same trend was observed for the AUC0-t.

Table 19 summarizes the relationship between genetic factors and CPF pharmacokinetics, genetic factors and clinical manifestations of CF. A correlation was found between polymorphic variants of genes associated with slow metabolism of xenobiotics and high values of studied pharmacokinetic parameters, especially with such an important predictor of the effectiveness of therapy as AUC. In addition, several qualitative features were identified as predictors that determine the inclusion of a patient in the group of “slow” metabolizers: less pathogenic microflora, intermittent inoculation of *P. aeruginosa*, liver damage. Furthermore, during the regression analysis, indirect data were obtained on the effect of age and the F508del/F508del genotype on the CPF pharmacokinetics.

## 4. Discussion

The results of our study demonstrate a high variability in the estimated pharmacokinetic parameters of CPF, which may be explained by a small number of study subjects in each of the age groups. With age, there was a tendency for longer Tmax. In the group of older children this fact may be associated with taking the tablet form of the drug, as well as with the formation of refluxes from the digestive tube against the background of the progression of the bronchopulmonary process.

Previously published studies of the pharmacokinetics of drugs in patients with CF found delayed absorption of oral dosage forms, an increase in the volume of distribution of some drugs, a trend towards a decrease in plasma concentrations of drugs, an acceleration of excretion [30].

Estimation of the pharmacokinetics of CPF in children with CF include a limited number of studies. Payen S et al. (2003) showed that CPF is excreted much more slowly in neonates compared to other age groups, with clearance in patients with CF approximately twice as high as in patients without this pathology [25]. According to Rubio TT (1997), an evaluation of the pharmacokinetics of CPF in children with CF suggested that doses of 30 mg/kg/day intravenously or 40 mg/kg/day orally should be used to achieve optimal therapeutic concentrations [23]. In a study by Schaefer HG (1996), a linear correlation was found between the clearance of CPF (l/h) and body weight, the average bioavailability was 61.8%, renal clearance was 11.4 l/h, plasma protein binding rate was 34%. The authors suggested the following dosing regimen for CPF in children with CF: for younger children (weight 14 to 28 kg), 28–20 mg/kg orally twice a day, for older children (weight 28 to 42 kg), from 20 to 15 mg/kg orally twice a day; when administered intravenously, doses of 15 to 10 mg/kg twice a day are sufficient [24]. Rajagopalan P (2003) stated that CF was defined as a covariate affecting the degree of absorption of CPF in children, the data obtained indicated a slight decrease in absorption, which may be important in the case of Pseudomonas aeruginosa, requiring higher MIC values compared to other pathogens [26].

Novoselova O.G. et al. studied the relationship between the presence of polymorphic variants of the genes of the 1st phase of xenobiotic biotransformation and the effectiveness of antibiotic therapy in patients with CF. An increase in the frequency of the CYP2C9*3 allele and the CA genotype was revealed in the group of patients with FEV1 ≥ 80% [31]. FEV1 is the best predictor of mortality in CF patients [32], and adequate antibiotic therapy promotes recovery of lung function in CF. Based on the data obtained in this study, the carriage of “slow” CYP2C9 genotypes: R144C (rs1799853) and I359L (rs1057910), increases the therapeutic efficacy of antibacterial therapy.

Our results demonstrated the relationship between the CA genotype of the CYP2C9*3 variant (c.1075A > C) and the pharmacokinetic parameters of CPF. The presence of the CA genotype was associated with the highest values of AUC, but at the same time with lower values of Cmax compared with the AA genotype. Data published in other works indicate the existence of situations where the values of AUC and Cmax change in opposite directions. In such cases, it is proposed to focus on the indicator that is most closely related to the clinical efficacy of the drug. As is known, fluoroquinolones are concentration-dependent antimicrobial drugs with a persistent post-antibiotic effect. The clinical effect is determined by the ratio between the AUC and the MIC and the ratio between the Cmax and the MIC. Studies have shown that in settings with a high risk of developing antibiotic resistance, it is the AUC/MIC ratio that more accurately reflects the effectiveness of fluoroquinolones, and the AUC/MIC ratio > 125 indicates better therapeutic outcomes [33,34,35]. Previously published data indicated that CPF is an inhibitor of cytochrome P450 1A2 (CYP1A2) [36]. In our study, no genetic variants of the gene encoding this enzyme were determined in patients. Recently, data have appeared that many enzymes from the P450 family of monooxygenases involved in the metabolism of endogenous substrates are not as specific as previously thought [37]. Considering these data, the oxidation of the ciprofloxacin molecule is possible with the participation of the CYP2C9*3 enzyme. In patients with the AA genotype of the CYP2C9 * 3 gene I359L (c.1075A > C), an insufficient clinical effect is possible when using CPF. Our results indicate that the presence of slow CYP2D6*4 alleles is more common in the group of patients without meconium ileus; the development of this syndrome may be directly related to the presence of functional alleles of the CYP2D6*4 gene.

According to the published data, CYP2D6 expression in liver samples of newborns less than 7 days of age was higher than in first and second trimester fetal liver samples, but not significantly higher than in third trimester fetal samples. The activity of the CYP2D6 isoenzyme in infants reaches the level of adults [38]. In addition, data have been published that “fast” alleles of the CYP2D6 gene are more common in patients with inflammatory bowel diseases [39]. It is likely that the ontogeny of the CYP2D6 gene contributes to the development of meconium ileus in patients with CF.

An analysis of our data suggests a relationship between liver damage and the AG genotype of the CYP2D6*4 gene. CYP2D6 is involved in the metabolism of the endogenous cannabinoid anandamide with the formation of a potent selective cannabinoid CB2 receptor agonist, the activity of which is associated with antifibrogenic processes in the liver. The presence of a slow CYP2D6 genotype may contribute to a faster rate of fibrogenesis in the liver [40].

We found higher values of Cmax_norm of CPF in the group of patients with “slow” alleles. However, the result may be affected by uneven age distribution in the subgroups, which probably explains the trend towards higher AUC values in the GG group CYP2D6*4 gene.

In our work, the relationship between the F508del/F508del genotype and intermittent and chronic *P. aeruginosa* infection in CF patients was obtained. The F508del genotype belongs to the “severe” class of mutations in the *CFTR* gene, it specifies that the genotype is strongly correlated with pancreatic function, both in pediatric age and in adulthood [41,42]. The “severe” genotype in CF is associated with earlier development and a high incidence of chronic Gram-negative respiratory tract infection compared to the “mild” genotypes [43,44]. However, according to the Register of CF patients in Russia and the European CF Society Patient Registry, no significant relationship between microbiological diagnosis and genotype homozygotes for the F508del was found [44,45]. It was also shown that the group of homozygotes for the F508del variant is characterized by higher concentrations of CPF after 1.5, 3.0 and 7.5 h and a higher value of Cmax_norm. In patients with CF, exocrine pancreatic insufficiency is common, with decreased bicarbonate secretion leading to duodenal acidification. CPF has a maximum solubility at a pH below 5 and a minimum near the isoelectric point (pH 7). After oral administration, CPF is rapidly absorbed from the gastrointestinal tract by passive diffusion and reaches its maximum serum concentration within 2 h. The rate of absorption is influenced by intestinal pH, with absorption in the duodenum and proximal jejunum greater than in the distal small intestine [46]. It is generally accepted that Tmax or Cmax is more indicative of the rate of absorption [47]. In this regard, the recorded high plasma concentrations of CPF may be associated with a more intense rate of its absorption in patients from the group of homozygotes for F508del due to the most pronounced acidification of the gastrointestinal tract in patients with “severe” genotypes compared to “mild” genotypes.

A correlation was found between the presence of “slow” NAT2 alleles (341T > C) and higher Cmax_norm values; however, Cmax and AUC0-t_norm were higher in the TT genotype group, into which there was a greater extent of patients with the F508del/F508del genotype. The predictors included in the logit-regression equation with the dependent variable “NAT2 gene (341T > C)” do not fully explain the differences between the groups of “fast” and “slow” alleles and identify logically substantiated relationships between these genotypes and CPF pharmacokinetics. The presence in the majority of patients with the F508del/F508del genotype in the group of “fast” alleles of the NAT2 gene (341T > C) explains the high values of AUC0-t_norm, which is comparable with the results of Cmax_norm in homozygotes for the F508del genetic variant. Patients with the F508del/F508del genotype were more common in the group of children with the AA genotype of the *GSTP1* gene, which indirectly explains the high Cmax_norm in this group. Such an important predictor as AUC0-t_norm was higher in the group with “slow” genotypes of the *GSTP1* gene, which is associated with the presence of intermittent *P. aeruginosa* seeding in this subgroup, while the AA genotype is associated with chronic *P. aeruginosa* infection. The fact that the normal AA genotype was characterized by higher AUC0-t values, prolonged Tmax and high weight values is probably due to the predominance of older children in the group.

The *GCLC* gene is a genetic modifier of lung disease in CF [48]. McKone et al. described a significant effect of polymorphisms of the *GCLC*, TNR and GAG genes on lung function in CF only in patients with the “mild” genotype of the *CFTR* gene. This is due to the fact that one of the functions of the *CFTR* protein is to ensure the transport of glutathione into the bronchial secretion. Patients with a “severe” *CFTR* genotype have very low levels (<3%) of functioning *CFTR* protein, resulting in a more severe CF phenotype. Patients with a “mild” *CFTR* gene genotype have a milder clinical phenotype, which is most likely associated with the expression of an increased amount (5–13%) of the functioning *CFTR* protein. Since *CFTR* influences the transport of GSH into bronchial secretions, CF patients with GCLC gene variants associated with greater glutathione production, with class 3–5 mutations, should have higher levels of bronchial secretion glutathione, which is involved in protecting the lungs from oxidant-induced damage. Conversely, a pronounced decrease in the functional *CFTR* protein, associated with the presence of “severe” mutations in the patient, will cause low levels of glutathione in the bronchial secretion, regardless of the effect of polymorphisms of the *GCLC, GAG, TNR* genes on the synthesis of glutathione [48]. GCLC gene polymorphisms 7/7, GAG, TNR were associated with lower glutathione production compared to 8 GAG TNR and 9 GAG TNR [49,50].

Our work revealed a significant relationship between the GCLC 7/7 genotype and a high concentration of CPF at point 5 (7.5 h after the start of the drug). When conducting a logistic regression analysis, such qualitative indicators as intermittent and chronic *P. aeruginosa* infection and chronic staphylococcal infection were taken into account. We obtained an association of chronic staphylococcal infection with the GCLC 7/7 genotype, while *P. aeruginosa* was not included as a predictor in any of the equations. This may be due to the fact that higher concentrations of CPF in patients with “slow” alleles contribute to an increase in its therapeutic efficacy, and also due to the fact that more patients with “severe” *CFTR* genotypes were allocated to the group with the GCLC 7/7 genotype.

## 5. Conclusions

The results of our work indicate a possible relationship between the CA genotype of the CYP2C9 gene (c.1075A > C), the GG genotype of the CYP2D6*4 gene (1846G > A), the AG genotype of the GSTP1 gene (c.313A > G), the GCLC genotype 7/7 and the high values of the main predictor of the effectiveness of fluoroquinolones-AUC. Furthermore, the influence of the *CFTR* gene genotype on the CPF PK parameters cannot be excluded. The presence of “slow” genotypes contributes to the creation of therapeutic concentrations of CPF in the blood of children with CF, which may contribute to a more favorable course of the disease.

## Figures and Tables

**Figure 1 biomedicines-10-01050-f001:**
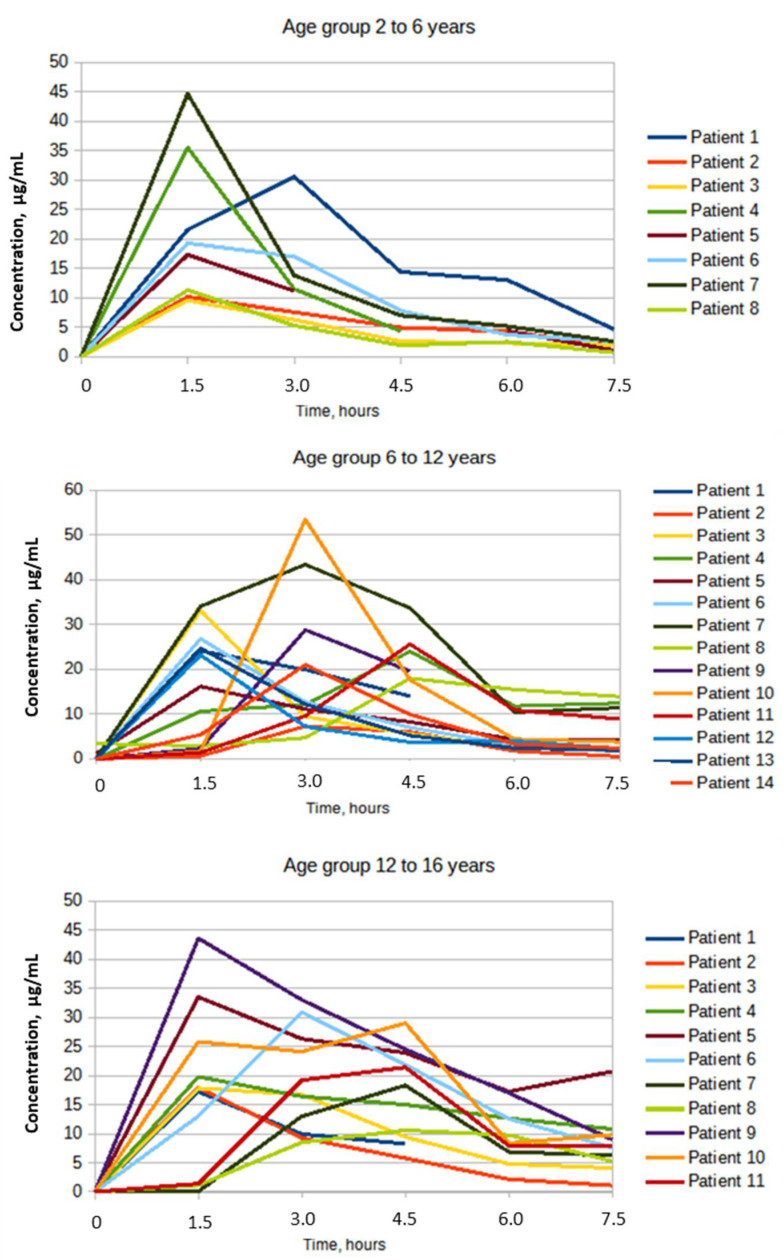
The dynamics of the CPF plasma concentration in different age groups measured within 7.5 h after a single dose.

**Table 1 biomedicines-10-01050-t001:** The basic characteristics of the study population.

Characteristics		Age Groups			
		2–6 Years	6–12 Years	12–18 Years	Total
Age, years	N	8	14	11	33
	Mean (±SD)	3.37 (0.91)	8.14 (8.00)	14.27 (1.61)	9.03 (4.48)
	Median	3.00	8.00	14.00	9.00
Weight, kg	Mean (±SD)	15.37 (3.19)	26.82 (8.75)	46.95 (9.69)	30.75 (14.79)
	Median	15.60	25.25	50.00	26.00
Height, cm	Mean (±SD)	99.12 (10.06)	129.07 (10.76)	156.15 (12.56)	130.8 (24.32)
	Median	100.00	128.00	157.70	130.00
BMI, kg/m^2^	Mean (±SD)	15.52 (1.31)	15.74 (2.81)	19.02 (1.67)	16.78 (2.65)
	Median	15.25	14.80	18.50	16.40
BMI percentile	Mean (±SD)	50.83 (29.11)	38.95 (35.42)	49.07 (26.23)	45.08 (30.69)
	Median	46.00	30.10	18.50	36.90
Sex	Male, n (%)	3 (37.5%)	7 (50%)	2 (18.2%)	12 (36.4%)

**Table 2 biomedicines-10-01050-t002:** Pharmacokinetic parameters of CPF in different age groups.

Parameter	2–6 Years (*n* = 8)		6–12 Years (*n* = 14)		12–18 Years (*n* = 11)	
	Mean ± SD	CV (%)	Mean ± SD	CV (%)	Mean ± SD	CV (%)
Dose, mg/kg	21.40 ± 3.38	-------	21.97 ± 3.65	---------	21.12 ± 2.21	---------
C_max_, µg/mL	22.32 ± 13.15	58.9	26.4 ± 11.33	42.89	23.63 ± 9.51	40.24
T_max_, hours	1.68 ± 0.53	31.42	2.67 ± 1.2	44.89	2.72 ± 1.47	53.99
AUC_0-t_ µg·h/mL	69.84 ± 35.29	50.54	85.44 ± 38.55	45.13	100.53 ± 46.6	46.31
AUC_0-t_norm_ (µg·h/mL)/(mg/kg)	3.33 ± 1.98	59.54	3.880 ± 1.46	37.71	4.77 ± 2.20	46.18
C_max_norm_ (µg/mL)/(mg/kg)	1.03 ± 0.545	52.98	1.1886 ± 0.40	33.47	1.12 ± 0.47	41.70

**Table 3 biomedicines-10-01050-t003:** Characteristics of patients with different genotypes of cytochrome P450 monooxygenase enzymes, glutathione S-transferases and N-acetyltransferases.

	Genotypes		*n*	Age, Years Mean (±SD)	Male, *n*
Monooxygenases of the CYP450 family	CYP2C9*3 (c.1075A > C)	AA	25	8.24 (4.22)	9
		AC	6	9.88 (5.38)	5
		CC	1	12	0
	CYP2C9*2 (c.430C > T)	CC	27	9.1 (4.6)	10
		CT	5	6.8 (3.6)	4
	CYP2C19*2 (c.681G > A)	GG	22	8.4 (4.5)	8
		GA	9	9.4 (4.8)	5
		AA	1	10.0	1
	CYP2D6*4 (1846G > A)	GG	19	8.9 (4.3)	6
		GA	12	8.64 (4.9)	7
		AA	1	6.0	1
Glutathione-S-transferase	GSTT1	N/N	26	9.4 (4.5)	9
		D/D	6	6.7 (2.6)	4
	GSTMI	Without deletion	14	9.5 (4.8)	9
		Genotype Zero	18	8.5 (4.1)	4
	GSTP1	AA	15	8.2 (4.9)	5
		AG	14	9 (4.3)	8
		GG	2	9 (5.7)	1
N-acetyltransferase	NAT2 (282C > T)	CC	18	8.16 (4.29)	8
		CT	13	9.47 (4.72)	6
	NAT2 (590G > A)	GG	21	9 (4.44)	9
		GA	11	8.31 (4.64)	5
	NAT2 (341T > C)	TT	7	9.0 (3.40)	2
		TC	14	10.13 (4.78)	7
		CC	11	6.83 (3.66)	5
	NAT2 481C > T	CC	8	9.4 (4.70)	3
		CT	14	9.79 (4.39)	6
		TT	10	6.6 (3.98)	5
	NAT2 803A> G	AA	7	8.67 (4.85)	5
		AG	13	10.38 (4.29)	5
		GG	11	6.82 (4.14)	4

**Table 4 biomedicines-10-01050-t004:** Pharmacokinetic parameters of CPF depending on CYP2C9*3 gene polymorphism (I359L, c.1075A > C).

Parameter	Statistical Indicators	Genotype		
		AA	CA	CC
Dose, mg/kg	N	25	7	1
	Mean (±SD)	21.51 (3.46)	19.14 (8.76)	20
	Median (75%Q3–25%Q1)	20 (24–19.5)	21.7 (24.5–18.8)	20
	CV	16.1	45.76	-
Concentration at point 1, µg/mL	N	25	6	1
	Mean (±SD)	18.72 (13.17)	16.19 (10.31)	1.4
	Median (75%Q3–25%Q1)	17.89 (25.8–9.58)	15.16 (19.79–10.69)	1.4
	CV	70.33	63.69	-
Concentration at point 2, µg/mL	N	25	6	1
	Mean (±SD)	17.37 (12.19)	16.93 (9.89)	19.2
	Median (75%Q3–25%Q1)	12.7 (21–9.43)	14.26 (26.3–11.17)	19.2
	CV	70.20	58.40	-
Concentration at point 3, µg/mL	N	25	5	1
	Mean (±SD)	11.78 (8.83)	20.55 (3.94)	21.4
	Median (75%Q3–25%Q1)	8.26 (17.6–5.56)	21.85 (23.89–18.0)	21.4
	CV	75.04	19.17	-
Concentration at point 4, µg/mL	N	14	4	1
	Mean (±SD)	5.16 (4.14)	11.27 (4.78)	7.92
	Median (75%Q3–25%Q1)	4.0 (5.16–2.6)	12.6 (14.06–8.48)	7.92
	CV	80.12	42.39	-
Concentration at point 5, µg/mL	N	25	6	1
	Mean (±SD)	4.18 (3.14)	11.08 (6.55)	7.8
	Median (75%Q3–25%Q1)	2.6 (5.96–1.84)	11.6 (13.9–7.62)	7.8
	CV	75.24	59.14	-
AUC_0-t_, µg·h/mL	N	25	6	1
	Mean (±SD)	84.13 (42.61)	104.59 (37.47)	80.73
	Median (75%Q3–25%Q1)	75.62 (107.94–61.17)	100.51 (123.20–74.60)	80.73
	CV	50.64	35.83	-
C_max_, µg/mL	N	25	6	1
	Mean (±SD)	25.31 (11.94)	23.92 (6.88)	21.4
	Median (75%Q3–25%Q1)	24.09 (30.61–17.81)	21.90 (30.9–18)	21.4
	CV	47.17	28.75	-
T_max_, h	N	25	6	1
	Mean (±SD)	2.22 (1.07)	2.75 (1.48)	4.5
	Median (75%Q3–25%Q1)	1.5 (3.0–1.5)	2.25 (4.5–1.5)	4.5
	CV	48.25	53.63	-
AUC_0-t_norm_, µg·h/mL	N	25	6	one
	Mean (±SD)	3.95 (2.02)	4.69 (1.49)	4.04
	Median (75%Q3–25%Q1)	3.31 (4.56–2.89)	4.59 (5.68–3.97)	4.04
	CV	51.15	31.78	-
C_max_norn_, µg/mL	N	25	6	1
	Mean (±SD)	1.17 (0.48)	1.08 (0.30)	1.07
	Median (75%Q3–25%Q1)	1.05 (1.45–0.83)	1.08 (1.37–0.82)	1.07
	CV	41.51	27.71	-

**Table 5 biomedicines-10-01050-t005:** Comparative data of group mean concentrations of CPF at point 5 for the grouping trait “CYP2C9*3 polymorphism I359L(c.1075A > C)” in patients with genotypes AA, CA, CC.

Quantitative Characteristic Name	Mean Value in the Group/Number of Observations			The Level of Statistical Significance, *p* ≤ 0.05	
						Kruskal-Wallis Test	Van der Waerden Test	AA	CA	CC
CPF concentration at point 5	4.18/25	11.08/6	7.80/1	*p* = 0.05	*p* = 0.04

**Table 6 biomedicines-10-01050-t006:** The result of discriminant analysis with the dependent variable “CYP2C9*3 gene polymorphism I359L(c.1075A > C)” and pharmacokinetic parameters of CPF.

*n* = 30	The Result of the Analysis of the Discriminant Function Step 2, N Variables in Model l: 2, Grouping: CYP2C9*3 I359L(c.1075A > C) (2 Groups) Wilks’ Lambda: 0.42029 Approximation. F(2.27) = 18.621, *p* < 0.0000					
	Wilks’ Lambda	Partial Lambda	F-Remove (1.27)	*p*-Value	Toler.	1-Toler. (R-Sqr.)
CPF concentration at point 5	0.857958	0.489875	28.11609	0.000014	0.380098	0.619902
CPF concentration at point 3	0.500211	0.840231	5.13404	0.031690	0.380098	0.619902

**Table 7 biomedicines-10-01050-t007:** Evaluation of logistic regression parameters with the dependent variable “CYP2C9*3 gene polymorphism I359L (c.1075A > C)”.

Predictor	Regression Coefficient	Standardized Regression Coefficient	Wald Chi^2^	The Level of Statistical Significance, *p*	Standard Error
Free member 1 (CA genotype)	3.3573		5.7513	0.0165	1.4000
C_max_, µg/mL	0.4371	2.6625	4.6479	0.0311	0.2027
AUC, µg·h/mL	−0.0900	−2.0260	4.1340	0.0420	0.0442
Total dose of CPF, mg	−0.0050	−0.7412	4.2320	0.0397	0.0024
Chronic staphylococcal infection	3.9051	0.9151	6.3188	0.0119	1.5535

**Table 8 biomedicines-10-01050-t008:** Pharmacokinetic parameters of CPF in patients with various genotypes in patients with different genotype variants: GG, GA, AA of the CYP2D6*4 gene (1846G > A).

Indicator	Meaning	GG	GA	AA
Dose, mg/kg	*n*	19	13	1
	Mean (±SD)	21.32 (3.13)	20.01 (6.81)	26.7
	Median	20.0 (24.5–19.4)	21.7 (22.8–19.6)	26.7
	(75%Q3–25%Q1)			
	CV	14.67	34.05	-
Concentration at point 1, µg/mL	*n*	19	12	1
	Mean (±SD)	15.25 (12.51)	20.84 (13.03)	26.8
	Median (75%Q3–25%Q1)	17.31 (24.09–2.30)	18,51 (33.60–10.96)	26.8
	CV	82.03	62.52	-
Concentration at point 2, µg/mL	*n*	19	12	1
	Mean (±SD)	17.16 (11.07)	18.01 (12.89)	12.7
	Median (75%Q3–25%Q1)	13.8 (21.0–9.6)	11.81 (30.76–8.37)	12.7
	CV	64.51	71.55	-
Concentration at point 3, µg/mL	*n*	19	12	1
	Mean (±SD)	13.86 (7.98)	13.51 (10.43)	7.00
	Median (75%Q3–25%Q1)	11.94 (19.6–7.0)	11.34 (22.93–4.95)	7.00
	CV	57.56	77.14	-
Concentration at point 4, µg/mL	*n*	13	5	1
	Mean (±SD)	5.83 (3.74)	9.31(6.77)	3.0
	Median	4.40 (6.88–3.72)	12.56 (12.64–2.60)	3.0
	(75%Q3–25%Q1)			
CV	64.26	72.72	-
Concentration at point 5, µg/mL	*n*	19	12	1
	Mean (±SD)	5.71 (5.03)	5.71 (4.44)	1.7
	Median (75%Q3–25%Q1)	4.09 (8.55–2.30)	5.32 (9.89–1.69)	1.7
	CV	88.06	77.78	-
AUC_0-t_, µg·h/mL	*n*	19	12	1
	Mean (±SD)	83.89 (31.87)	95.18 (55.02)	75.53
	Median (75%Q3–25%Q1)	74.59 (107.94–61.79)	89.96 (125.33–50.15)	75.53
	CV	37.88	57.81	-
C_max_, µg/mL	*n*	19	12	1
	Mean (±SD)	24.45 (10.32)	25.53 (12.62)	26.8
	Median (75%Q3–25%Q1)	21.4 (28.8–17.89)	27.31 (34.33–14.26)	26.8
	CV	42.20	49.42	-
T_max_, hours	*n*	19	12	1
	Mean (±SD)	2.53 (1.33)	2.25 (1.01)	1.5
	Median	1.5 (4.5–1.5)	1.5 (3.0–1.5)	1.5
	(75%Q3–25%Q1)			
	CV	52.56	44.94	-
AUC_0-t_norm_, (µg·h/mL)/(mg/kg)	*n*	19	12	1
	Mean (±SD)	3.96 (1.42)	4.41 (2.56)	2.83
	Median (75%Q3–25%Q1)	3.68 (4.56–3.06)	3.86 (6.15–2.48)	2.83
	CV	35.89	58.16	-
C_max_norm_, (µg/mL)/(mg/kg)	*n*	19	12	1
	Mean (±SD)	1.13 (0.37)	1.18 (0.57)	1.00
	Median	1.05 (1.38–0.94)	1.31 (1.48–0.68)	1.00
	(75%Q3–25%Q1)			
	CV	32.47	48.49	-

**Table 9 biomedicines-10-01050-t009:** Evaluation of logistic regression parameters in patients with different genotype variants: GG, GA, AA of the CYP2D6*4 gene (1846G > A).

Predictor	Regression Coefficient	Standardized Regression Coefficient	Wald Chi^2^	The Level of Statistical Significance, *p*	Standard Error
Free member 0 (GG genotype)	−1495.1	860.1	3.0219	0.0821	NA
Free member 1 (GA genotype)	−1428.2	821.7	3.0212	0.0822	NA
Age, years	25.6330	15.2937	2.8091	0.0937	59.9328
Weight, kg	25.2035	14.7289	2.9280	0.0871	187.1
Height, cm	6.5305	3.7882	2.9719	0.0847	83.1126
BMI, kg/m^2^	19.7351	11.4131	2.9900	0.0838	27.7247
Dose of CPF, mg/kg	35.1704	20.2638	3.0124	0.0826	63.7317
AUC, µg·h/mL	1.8107	1.0713	2.8565	0.0910	40.7737
Total dose of CPF, mg	−2.2508	1.3090	2.9566	0.0855	−333.5
C_max_norm_ (µg/mL)/(mg/kg)	−90.7806	53.5454	2.8744	0.0900	−22.4050
Anamnesis of meconium ileus	133.0	80.4820	2.7325	0.0983	24.9956
Liver damage without cirrhosis (elevation of liver enzymes plus ultrasound diagnostics)	−71.2019	42.1590	2.8524	0.0912	−37.3079

**Table 10 biomedicines-10-01050-t010:** Pharmacokinetic parameters of ciprofloxacin in children with CF with different types of the GSTP1 gene polymorphism.

Parameter	Statistical Indicators	AA	AG	GG
Dose, mg/kg	*n*	15	15	2
	Mean (±SD)	21.03 (3.37)	21.03 (3.37)	24.35 (3.75)
	Median (75%Q3–25%Q1)	20.0 (22.8–19.4)	20.0 (22.8–19.4)	24.35 (27.0–21.70)
	CV	16.01	16.01	15.39
Concentration at point 1, µg/mL	*n*	15	15	2
	Mean (±SD)	14.91 (10.66)	14.91 (10.66)	28.86 (22.40)
	Median (75%Q3–25%Q1)	16.18 (21.57–5.40)	16.18 (21.57–5.40)	28.86 (44.7–13.02)
	CV	71.52	71.52	77.63
Concentration at point 2, µg/mL	*n*	15	15	2
	Mean (±SD)	19.46 (14.30)	19.46 (14.30)	22.35 (12.09)
	Median (75%Q3–25%Q1)	12.98 (28.80–9.43)	12.98 (28.80–9.43)	22.35 (30.90–13.80)
	CV	73.50	73.50	54.10
Concentration at point 3, µg/mL	*n*	14	14	2
	Mean (±SD)	13.11 (9.60)	13.11 (9.60)	14.42 (10.50)
	Median (75%Q3–25%Q1)	9.67 (8.30–5.56)	9.67 (8.30–5.56)	14.42 (21.85–7.0)
	CV	73.24	73.24	72.78
Concentration at point 4, µg/mL	*n*	7	7	2
	Mean (±SD)	4.38 (1.34)	4.38 (1.34)	8.86 (5.23)
	Median (75%Q3–25%Q1)	4.4 (4.8–3.28)	4.4 (4.8–3.28)	8.86 (12.56–5.16)
	CV	30.67	30.67	59.06
Concentration at point 5, µg/mL	*n*	15	15	2
	Mean (±SD)	4.31 (3.08)	4.31 (3.08)	5.11 (3.55)
	Median (75%Q3–25%Q1)	4.09 (5.96–1.84)	4.09 (5.96–1.84)	5.11 (7.62–2.60)
	CV	71.33	71.33	69.47
AUC_0-t_, µg·h/mL	*n*	15	15	2
	Mean(±SD)	83.25 (44.05)	83.25 (44.05)	115.57 (10.79)
	Median (75%Q3–25%Q1)	68.25 (118.65–61.17)	68.25 (118.65–61.17)	115.56 (123.20–107.94)
	CV	52.92	52.92	9.34
C_max_, µg/mL	*n*	15	15	2
	Mean(±SD)	23.82 (12.57)	23.82 (12.57)	37.8 (9.76)
	Median (75%Q3–25%Q1)	18.30 (30.6–16.18)	18.30 (30.6–16.18)	37.80 (44.70–30.90)
	CV	52.78	52.78	25.82
T_max_, hours	*n*	15	15	2
	Mean(±SD)	2.4 (1.11)	2.4 (1.11)	2.25 (1.06)
	Median (75%Q3–25%Q1)	1.5 (3.0–1.5)	1.5 (3.0–1.5)	2.25 (3.0–1.5)
	CV	46.05	46.05	47.14
AUC_0-t_norm_, (µg·h/mL)/(mg/kg)	*n*	15	15	2
	Mean (±SD)	3.93 (1.89)	3.93 (1.89)	4.84 (1.19)
	Median (75%Q3–25%Q1)	3.27 (4.88–2.47)	3.27 (4.88–2.47)	4.84 (5.68–3.99)
	CV	48.17	48.17	24.55
C_max_norm_, (µg/mL)/(mg/kg)	*n*	15	15	2
	Mean (±SD)	1.11 (0.49)	1.11 (0.49)	1.54 (0.16)
	Median (75%Q3–25%Q1)	0.99 (1.45–0.69)	0.99 (1.45–0.69)	1.54 (1.66–1.42)
	CV	43.87	43.87	10.64

**Table 11 biomedicines-10-01050-t011:** Estimated logistic regression parameters for the dependent variable “GSTP1 gene polymorphism”.

Predictor	Regression Coefficient	Standardized Regression Coefficient	Wald Chi^2^	The Level of Statistical Significance, *p*	Standard Error
Free member 0 (genotype AA)	550.9	250.7	4.8289	0.0280	----------
Free member 1 (GA genotype)	634.3	286.4	4.9053	0.0268	----------
Weight, kg	1.4388	0.7370	3.8118	0.0509	10.3566
Dose of ciprofloxacin, mg/kg	−28.7268	12.9844	4.8947	0.0269	−52.7091
C_max_, µg/mL	−15.2313	7.9661	3.6559	0.0559	−94.2176
T_max_, hours	39.6197	17.8781	4.9111	0.0267	23.9541
AUC, µg·h/mL	8.7552	4.0921	4.5777	0.0324	200.5
AUC_0-t_norm,_ (µg·h/mL)/(mg/kg)	−247.3	114.5	4.6616	0.0308	−257.6
C_max_norm_, (µg/mL)/(mg/kg)	516.6	253 I	4.1410	0.0419	129.6
Use of azithromycin	−127.6	58.0950	4.8273	0.0280	−31.6519
Chronic persistent Pseudomonas aeruginosa infection	102.0	46.7224	4.7666	0.0290	21.3175
Chronic intermittent Pseudomonas aeruginosa infection	−87.2033	39.6337	4.8410	0.0278	−20.6822
Liver damage	40.9851	18.8952	4.7049	0.0301	21.7897
Pancreatic elastase	−108.1	49.2495	4.8144	0.0282	−20.5988
Homozygotes for F508del	148.0	66.7351	4.9194	0.0266	36.7042

**Table 12 biomedicines-10-01050-t012:** Modules of standardized regression coefficients of predictors for the dependent variable “GSTP1 gene polymorphism”.

Predictor	Modules of Standardized Regression Coefficients
AUC_0-t_norm_, (µg·h/mL)/(mg/kg)	|257.6|
AUC, µg·h/mL	|200.5|
C_max_norm_, (µg/mL)/(mg/kg)	|129.6|
C_max_, µg/mL	|94.22|
CPF dose, mg/kg	|52.71|
Homozygotes for delF508	|36.70|
Use of azithromycin	|31.65|
T_max_, hours	|23.95|
Liver damage	|21.79|
Chronic persistent Pseudomonas aeruginosa infection	|21.32|
Chronic intermittent Pseudomonas aeruginosa infection	|20.68|
Pancreatic elastase	|20.61|
Weight, kg	|10.36|

**Table 13 biomedicines-10-01050-t013:** Relationship between pharmacokinetic parameters, clinical signs and genotypes of the GSTP1 gene.

GSTP1 Genotype	Predictors	
	Pharmacokinetic Parameters	Clinical Signs
AA	High values of C_max_norm_ High AUC Longer T_max_	Fecal elastase level above 200 µg/g (in stool) Liver damage without cirrhosis Chronic Pseudomonas Infection Genotype F508del/F508del High weight and percentile BMI
GA	High AUC_0-t_norm_	Intermittent Pseudomonas Infection

**Table 14 biomedicines-10-01050-t014:** Pharmacokinetic parameters of ciprofloxacin in children and adolescents with different genotypes of the GCLC gene.

Parameter	Statistical Indicators	GCLC Genotypes	
		“Other”	“7/7” Genotype
Dose, mg/kg	*n*	18	14
	Mean (±SD)	20.04 (5.66)	22.1 (3.93)
	Median (75%Q3–25%Q1)	20.0 (22.8–19.5)	20.85 (25.0–19.40)
	CV	28.26	17.78
Concentration at point 1, µg/mL	*n*	17	14
	Mean (±SD)	17.92 (12.81)	17.79 (13.45)
	Median (75%Q3–25%Q1)	17.90 (25.80–9.58)	17.56 (24.09–5.40)
	CV	71.47	75.63
Concentration at point 2, µg/mL	*n*	17	14
	Mean (±SD)	15.35 (12.84)	18.80 (9.37)
	Median (75%Q3–25%Q1)	11.09 (17.0–7.31)	16.57 (21.0–12.13)
	CV	83.66	49.87
Concentration at point 3, µg/mL	*n*	17	13
	Mean (±SD)	10.11 (8.27)	17.29 (8.03)
	Median (75%Q3–25%Q1)	6.03 (17.6–4.96)	15.0 (23.89–9.90)
	CV	81.86	46.40
Concentration at point 4, µg/mL	*n*	10	8
	Mean (±SD)	5.68 (5.65)	6.99 (3.31)
	Median (75%Q3–25%Q1)	3.36 (4.40–2.40)	6.02 (9.40–4.60)
	CV	99.47	47.28
Concentration at point 5, µg/mL	*n*	17	14
	Mean (±SD)	3.73 (3.72)	7.69 (5.13)
	Median (75%Q3–25%Q1)	2.01 (4.03–1.70)	7.04 (10.79–4.09)
	CV	99.59	66.73
AUC_0-t_, µg·h/mL	*n*	17	14
	Mean (±SD)	76.22 (40.84)	99.48 (39.75)
	Median (75%Q3–25%Q1)	73.68 (82.75–53.48)	88.95 (110.34–68.25)
	CV	53.58	39.95
C_max_, µg/mL	*n*	17	14
	Mean(±SD)	24.0 (12.56)	25.63 (9.20)
	Median (75%Q3–25%Q1)	23.2 (29.0–16.18)	22.7 (30.61–18.30)
	CV	52.32	35.89
T_max_, hours	*n*	17	14
	Mean (±SD)	2.12 (1.07)	2.68 (1.34)
	Median (75%Q3–25%Q1)	1.5 (3.0–1.5)	2.25 (4.5–1.5)
	CV	50.45	49.98
AUC_0-t_norm_, (µg·h/mL)/(mg/kg)	*n*	17	14
	Mean (±SD)	3.62 (2.03)	4.55 (1.68)
	Median (75%Q3–25%Q1)	3.31	4.14
	CV	(3.97–2.25)	(6.30–3.15)
C_max_norm_, (µg/mL)/(mg/kg)	*n*	56.09	36.86
	Mean (±SD)	17	14
	Median (75%Q3–25%Q1)	1.12 (0.53)	1.16 (0.34)
	CV	1.00 (1.44–0.75)	1.06 (1.38–0.94)

**Table 15 biomedicines-10-01050-t015:** Estimated logistic regression parameters based on CPF concentration in patients with different genotype variants of GCLC gene.

Predictor	Regression Coefficient	Standardized Regression Coefficient	Wald Chi-Squared Test	The Level of Statistical Significance, *p*	Standard Error
“Other” genotypes	6.3620	NA	5.4146	0.0200	2.7341
CPF concentration at point 5	−0.5254	−1.3908	6.0159	0.0142	0.2142
Chronic staphylococcal infection	−4.2264	−0.9606	3.6440	0.0563	2.2140

**Table 16 biomedicines-10-01050-t016:** Estimated logistic regression parameters with exclusion of CPF concentration in patients with different genotype variants of GCLC gene.

Predictor	Regression Coefficient	Standardized Regression Coefficient	Wald Chi-Squared Test	The Level of Statistical Significance, *p*	Standard Error
Weight, kg	−0.4636	0.7370	3.8118	0.0509	10.3566
Height, cm	0.2174	0.0934	5.4219	0.0199	2.7521
Dose of ciprofloxacin, mg/kg	−0.4228	0.2168	3.8386	0.0501	−0.7830
T_max_, hours	−1.3875	0.7270	3.6426	0.0563	−0.8905
Chronic staphylococcal infection	−2.5359	1.5194	2.7859	0.0951	−0.6015

**Table 17 biomedicines-10-01050-t017:** Evaluation of logistic regression parameters in patients with different variants of the TT, TC, CC genotypes of the genetic variation of NAT2 (341T > C).

Predictor	Regression Coefficient	Standardized Regression Coefficient	Wald Chi-Squared Test	The Level of Statistical Significance, *p*	Standard Error
TT genotype	22.2390	NA	4.3622	0.0367	10.6479
TS genotype	25.0517	NA	5.2690	0.0217	10.9137
F508del/F508del genotype	2.4789	0.5809	4.6230	0.0315	1.1529
Concentration at point 2, µg/mL	−0.1596	−1.0190	5.2864	0.0215	0.0694
Concentration at point 3, µg/mL	0.4694	2.2766	3.9464	0.0470	0.2363
C_max_, µg/mL	1.7517	10.6082	6.4388	0.0112	0.6903
T_max_, hours	−1.9260	−1.2793	3.6026	0.0577	1.0147
AUC_0-t_, µg·h/mL	−0.3396	−7.7922	4.9033	0.0268	0.1534
AUC_0-t_norm_, (µg·h/mL)/(mg/kg)	6.7110	7.0541	4.4481	0.0349	3.1820
C_max_norm_, (µg/mL)/(mg/kg)	−38.2657	−1.3516	5.9664	0.0146	15.6659
Dose of CPF, mg/kg	−0.9643	−1.7227	3.8817	0.0488	0.4894

**Table 18 biomedicines-10-01050-t018:** Estimation of logistic regression parameters in patients with the F508del/F508del genotype and other genotypes.

Predictor	Regression Coefficient	Standardized Regression Coefficient	Wald Chi-Squared Test	The Level of Statistical Significance, *p*	Standard Error
Other genotypes	54.9688	NA	5.1170	0.0237	24.2957
Chronic and persistent Pseudomonas aeruginosa infection	−4.3270	−1.0495	3.8955	0.0484	2.1923
Concentration at point 1, µg/mL	−0.3704	−2.6617	3.4964	0.0615	0.1981
Concentration at point 2, µg/mL	−0.9071	−5.7526	4.9559	0.0260	0.4075
Concentration at point 5, µg/mL	−1.1133	−2.8468	3.6473	0.0562	0.5829
C_max_, µg/mL	2.4391	14.9399	6.0739	0.0137	0.9897
AUC_0-t_, µg·h/mL	0.3349	7.6692	4.4298	0.0353	0.1591
C_max_norm_, (µg·h/mL)/(mg/kg)	−48.1829	−11.9427	5.4855	0.0192	20.5723
Dose of CPF, mg/kg	−2.6767	−4.7333	5.2891	0.0215	1.1639

**Table 19 biomedicines-10-01050-t019:** The possible relationship between genetic factors, CPF pharmacokinetics and clinical manifestations of CF.

Genotype	Predictors	
	Pharmacokinetic Parameters	Clinical Manifestations
CYP2C9*3 (I359L, c.1075A > C)		
AA	High values of C_max_	-
CYP2D6*4 (1846G > A)		
GG	High values of AUC	History of meconium ileus A greater number of older children were here, thus weight, height and BMI were correspondingly higher
GCLC		
7/7 genotype	Longer T_max_	Chronic staphylococcal infection Higher weights compared to “other” genotypes
*GSTP1*		
AA	High values of C_max_norm_ High values of AUC Longer T_max_	Fecal elastase level above 200 µg/g Liver damage Chronic persistent Pseudomonas aeruginosa infection Genotype F508del/F508del High weight and percentile of BMI
GA	High values of AUC_0-t_norm_ High values of CPF dose, mg/kg	Intermittent Pseudomonas Infection Use of azithromycin
NAT2 (341T > C)		
TT	C_max_ AUC_0-t_norm_	Genotype F508del/F508del
TC CC	High values of CPF concentration after 1.5 h, 3.0 h, 7.5 h C_max_norm_ Longer T_max_	-
F508del/F508del		
Present	High concentrations of CPF after 1.5 h, 3.0 h and 7.5 h of taking the drug High values of C_max_norm_	Chronic Pseudomonas aerugenosae infection
Absent	C_max_	-

## Data Availability

https://mukoviscidoz.org/mukovistsidoz-v-rossii.html, accessed on 10 January 2022.

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
