# Peer review of "Gene Polymorphism of Biotransformation Enzymes and Ciprofloxacin Pharmacokinetics in Pediatric Patients with Cystic Fibrosis"

_biomedicines, 2022, doi:10.3390/biomedicines10051050_

Round 1
Reviewer 1 Report
The paper is well done, the results well explained and the topic of interest.
I recommend minimal changes:
- Introduction: specifies the number of CFTR2 disease-causing mutations;
- terms such as CFTR and Pseudomonas aeruginosa must be in italics;
- always uses CF and not "cystic fibrosis"
- also reports data from the European register and not only from the Russian register;
- interesting and useful to specify the results of the homozygous F508del: it specifies that the genotype is strongly correlated with pancreatic function, both in pediatric age (Terlizzi V, et al. Prediction of acute pancreatitis risk based on PIP score in children with cystic fibrosis. J Cyst Fibros. 2014 Sep; 13 (5): 579-84), both in adulthood (Ooi CY, et al. Type of CFTR mutation determines risk of pancreatitis in patients with cystic fibrosis. Gastroenterology. 2011 Jan; 140 (1) : 153-61). These references must be added in the text.
Author Response
The team of authors is grateful for the huge amount of work done!
We did make changes in the text according to your comments and add recommended citations in the Discussion section:
[Terlizzi V, Tosco A, Tomaiuolo R, Sepe A, Amato N, Casale A, Mercogliano C, De Gregorio F, Improta F, Elce A, Castaldo G, Raia V. Prediction of acute pancreatitis risk based on PIP score in children with cystic fibrosis. J Cyst Fibros. 2014 Sep;13(5):579-84. doi: 10.1016/j.jcf.2014.01.007. Epub 2014 Feb 11. PMID: 24525081]
and
[Ooi CY, Dorfman R, Cipolli M, Gonska T, Castellani C, Keenan K, Freedman SD, Zielenski J, Berthiaume Y, Corey M, Schibli S, Tullis E, Durie PR. Type of CFTR mutation determines risk of pancreatitis in patients with cystic fibrosis. Gastroenterology. 2011 Jan;140(1):153-61. doi: 10.1053/j.gastro.2010.09.046. Epub 2010 Nov 9. PMID: 20923678]
Reviewer 2 Report
Dear Authors,
thank you very much for having taken up this research topic. The aim of the study is very interesting.
However, the form of the presented article raises major objections and raises many doubts, for example:
- lines 34/36 - I propose to standardize the way of notation the incidence of CF
- line 38 - gen CFTR write in italics (CFTR)
- lines 63/64 - among liver diseases affecting the absorption of drugs, it is worth to mention intrahepatic cholestasis (citing the work of Drzymala-Czyz - DOI: 10.1016/j.dld.2021.06.034.)
- lines 189/191 and table 1- it is worth mentioning exactly how many patients were included in the study? 33 as it follows from the table or 57 as it follows from the text?
- What were the inclusion criteria, did everyone have pancreatic insufficiency, did they have liver disease, etc.
- Table 1 - in my opinion, it is not worth providing so many statistical parameters, since there were only 8 patients in the group
- Table 1 - sex male n (%) - I do not understand, since there were 8 children in the group and eight of them were boys, how was it calculated that they constituted 37.5%?
- The authors state: The results revealed high inter-individual variability in all studied pharmacokinetic parameters. Is it not due to a too-small sample group? Has the group size to be included in the study been calculated?
- Figure 1 - Why only the results from only 3 patients were shown (?) when there were 8 patients in the group.
- Table 3 - In the abstract and verse 108 it is stated that there were 33 patients, from table 3 it is shown that there were 34 patients? Could you explain it?
- Table 4 - it should be definitely improved - no information on what is between Concentration at point 3 μg / ml and Concentration at point 4 μg / ml (?), at the end of the table what parameter the results refer to was not specified; Authors can't write "one" instead of 1.
- Table 5 - the results are hardly readable, incomprehensible what 0.05 refers to and what refers to 0.04
- lines 256-265 - whether the influence of factors other than the genotype on the obtained results was checked (maybe the severity of the CFTR genotype, liver diseases, PI)
- Table 7 is insignificant, it may be worth describing the result in the text?
- lines 297-309 - the authors state: Means and medians had no clear differences. I do not understand why the authors use many different tests looking for differences that will not come out in such small groups anyway? Has the group size been calculated?
- In table 10 - the authors list liver damage - what does it mean? How is it diagnosed (ultrasound, ALT, ASP, cirrhosis?)
- Table 14 - Can predictors for the GG genotype be given? This group includes only 2 children (not scientific evidence, but random occurrence!)
- Table 15 - What do statistical indicators - 7/7 mean? It is not explained? The whole table is not readable
- Why do tables 16, 17, 18 etc. look different from the previous ones? It is worth standardizing
- Table 20 - in my opinion, Pharmacokinetic parameters and Clinical manifestations cannot be mentioned for patients with the genotype CYP2C9 * 3 (CC), CYP2D6 * 4 (AA), GSTP1 (GG) since these groups included one or two people!
- Discussion - since some of the results should be treated as random (especially if there were 1 or two patients in the group), the discussion should be corrected accordingly.
Author Response
Thank you! The team of authors is grateful for the huge amount of work done, our answers are in the attached file.

Round 2
Reviewer 2 Report
The introduced changes significantly improved the quality of the manuscript. I recommend accepting the article for publication.